# Combined In Silico, Ex Vivo, and In Vivo Assessment of L-17, a Thiadiazine Derivative with Putative Neuro- and Cardioprotective and Antidepressant Effects

**DOI:** 10.3390/ijms222413626

**Published:** 2021-12-20

**Authors:** Alexey Sarapultsev, Pavel Vassiliev, Daniil Grinchii, Alexander Kiss, Mojmir Mach, Jana Osacka, Alexandra Balloova, Ruslan Paliokha, Andrey Kochetkov, Larisa Sidorova, Petr Sarapultsev, Oleg Chupakhin, Maxim Rantsev, Alexander Spasov, Eliyahu Dremencov

**Affiliations:** 1Institute of Immunology and Physiology of the Ural Branch of RAS, Pervomayskaya 106, 620049 Ekaterinburg, Russia; p.sarapultsev@gmail.com; 2School of Medical Biology, South Ural State University, Lenina 76, 454080 Chelyabinsk, Russia; 3Department of Pharmacology and Bioinformatics, Volgograd State Medical University, Pavshikh Bortsov Square 1, 400131 Volgograd, Russia; pvassiliev@mail.ru (P.V.); akocha@mail.ru (A.K.); aspasov@mail.ru (A.S.); 4Center of Biosciences, Institute of Molecular Physiology and Genetics, Slovak Academy of Sciences, Dúbravská Cesta 9, 840 05 Bratislava, Slovakia; daniil.grinchii@savba.sk (D.G.); Ruslan.Paliokha@savba.sk (R.P.); 5Biomedical Research Center, Institute of Experimental Endocrinology, Slovak Academy of Sciences, Dúbravská Cesta 9, 845 05 Bratislava, Slovakia; Alexander.Kiss@savba.sk (A.K.); Jana.Bundzikova@savba.sk (J.O.); 6Center of Experimental Medicine, Institute of Experimental Pharmacology and Toxicology, Slovak Academy of Sciences, Dúbravská Cesta 9, 840 04 Bratislava, Slovakia; mojmir.mach@savba.sk (M.M.); Alexandra.Balloova@savba.sk (A.B.); 7Ural Federal University named after the First President of Russia B. N. Yeltsin, 19 Mira Street, 620002 Ekaterinburg, Russia; vlapp@isnet.ru (L.S.); chupakhin@ios.uran.ru (O.C.); 8The IJ Postovsky Institute of Organic Synthesis of the Ural Branch of RAS, Akademicheskaya/S. Kovalevskoi, 22/20, 620990 Ekaterinburg, Russia; 9Ural State Medical University, Repina 3, 620014 Ekaterinburg, Russia; r-ma@bk.ru

**Keywords:** treatment-resistant depression, depression due to general medical condition, post-stroke depression, post-myocardial infarction (MI) depression, thiadizines, serotonin transporter (SERT), serotonin receptors 5-HT_3_ and 5-HT_1A_, docking energy, binding affinity, binding mechanism, c-*Fos* immunohistochemistry, electrophysiology in vivo

## Abstract

Depression associated with poor general medical condition, such as post-stroke (PSD) or post-myocardial infarction (PMID) depression, is characterized by resistance to classical antidepressants. Special treatment strategies should thus be developed for these conditions. Our study aims to investigate the mechanism of action of 2-morpholino-5-phenyl-6H-1,3,4-thiadiazine, hydrobromide (L-17), a recently designed thiadiazine derivative with putative neuro- and cardioprotective and antidepressant-like effects, using combined in silico (for prediction of the molecular binding mechanisms), ex vivo (for assessment of the neural excitability using c-*Fos* immunocytochemistry), and in vivo (for direct examination of the neuronal excitability) methodological approaches. We found that the predicted binding affinities of L-17 to serotonin (5-HT) transporter (SERT) and 5-HT_3_ and 5-HT_1A_ receptors are compatible with selective 5-HT serotonin reuptake inhibitors (SSRIs) and antagonists of 5-HT_3_ and 5-HT_1A_ receptors, respectively. L-17 robustly increased c-*Fos* immunoreactivity in the amygdala and decreased it in the hippocampus. L-17 dose-dependently inhibited 5-HT neurons of the dorsal raphe nucleus; this inhibition was partially reversed by the 5-HT_1A_ antagonist WAY100135. We suggest that L-17 is a potent 5-HT reuptake inhibitor and partial antagonist of 5-HT_3_ and 5-HT_1A_ receptors; the effects of L-17 on amygdaloid and hippocampal excitability might be mediated via 5-HT, and putatively mediate the antidepressant-like effects of this drug. Since L-17 also possesses neuro- and cardioprotective properties, it can be beneficial in PSD and PMID. Combined in silico predictions with ex vivo neurochemical and in vivo electrophysiological assessments might be a useful strategy for early assessment of the efficacy and neural mechanism of action of novel CNS drugs.

## 1. Introduction

L-17 (2-morpholino-5-phenyl-6H-1,3,4-thiadiazine, hydrobromide; Figure 1) is a thiadiazine derivative, synthesized by cyclocondensation of α-bromoacetophenone with the original morpholine-4-carbothionic acid hydrazide [1].

In our previous studies, we have shown, using animal models, a putative therapeutic effect of L-17 in myocardial infarction (MI) [2,3,4] and pancreatitis [5]. In these models, L-17 significantly decreased the initial infection area and accelerated the granulocytotic and suppressed cytokine component of the inflammatory reaction in rats after coagulation of the left coronary artery [2,3]. L-17 was also reported to attenuate the immune system response to immobilization stress in rats, suggesting an immunostimulatory effect of this compound during stress copying [4]. In addition, neuroprotective [1] and antidepressant-like [6] effects of L-17 in rats have been demonstrated. In this study, we aimed to investigate the mechanism of the putative beneficial effect of L-17 on the central nervous system (CNS), using a combination of in silico, Ex Vivo, and In Vivo methods [7].

For in silico assessments, we constructed a three-dimensional (3D) model of L-17; examined its interaction with different targets, such as serotonin (5-HT) transporter (SERT) and 5-HT receptors 3 and 1A (5-HT_3_/5-HT_1A_); and calculated the minimum binding energy. SERT and 5-HT_3_ were chosen because they play a primary role in the pathophysiology and treatment of CNS disorders.

Immunohistochemical assessment of the proto-oncogene *Fos* across the brain is an Ex Vivo experimental technique used for the investigation of the mechanism of action of novel CNS drugs. The activation of the proto-oncogene *Fos*, resulting in an increased expression of its protein product c-*Fos*, is a well-established marker for neuronal excitability within a specific brain area [8,9,10,11]. Increased proto-oncogene *Fos* expression in the amygdala, indicating increased neural excitability in this brain area, has been observed after administration of various CNS drugs, such as typical [12] and atypical antipsychotics [12,13,14], tricyclic antidepressants [13], and selective serotonin reuptake inhibitors (SSRIs) [13,15,16]. The SSRI citalopram-induced increase in c-*Fos* immunoreactivity was shown to be potentiated by the selective antagonism of 5-HT_1A_ receptors with WAY100638 [17], suggesting a key role of the 5-HT system in the modulation of amygdaloid excitability by CNS drugs. With regard to the prefrontal cortex (PFC), some SSRIs, such as fluvoxamine [16], and some atypical antipsychotics, such as olanzapine [18,19], increased local c-*Fos* immunoreactivity. On the other hand, 5-HT depletion led to an increased c-*Fos* immunoreactivity in the PFC and hippocampus [20], suggesting that extracellular 5-HT suppresses neural excitability and *Fos* proto-oncogene expression in these brain areas.

Finally, we used single-unit extracellular electrophysiology In Vivo to examine the effect of L-17 on the excitability of 5-HT neurons in the dorsal raphe nucleus (DRN) of rats. Multiple antidepressant drugs, such as tricyclic antidepressants, SSRIs, dual 5-HT/norepinephrine, triple 5-HT/norepinephrine/dopamine reuptake inhibitors, pyridoindoles (experimental drugs with putative triple reuptake inhibition property), and some atypical antipsychotics, exhibit a potent acute inhibitory effect on the excitability of 5-HT neurons of the DRN. This inhibition is usually reversed by the selective 5-HT_1A_ receptors antagonist, suggesting the involvement of the extracellular 5-HT. The assessment of the effect of an experimental CNS drug on the excitability of 5-HT neurons is a key marker for the preclinical assessment of its efficacy.

## 2. Results

### 2.1. In Silico Predictions

The following pharmacological activities were predicted for L-17 using PASS 10.4 Professional Extended software [21]: cognition disorders treatment (Pa = 0.489, Pa/Pi = 27.17), phobic disorders treatment (Pa = 0.703, Pa/Pi = 9.37), psychotropic (Pa = 0.258, P/Pi = 1.74), immunostimulant (Pa = 0.191, Pa/Pi = 1.39), and antidepressant (Pa = 0.162, Pa/Pi = 1.14). According to joint predictive evaluations in PASS and experimental data [1,6], the most likely targeted activities corresponding to the serotoninergic effects of compound L-17 were suggested to be: 5-HT_3_ antagonism (Pa = 0.139, Pa/Pi = 1.17), 5-HT release inhibition (Pa = 0.225, Pa/Pi = 1.01), and 5-HT reuptake blockade (Pa = 0.470, Pa/Pi = 6.35). While the L-17 compound manifests as an atypical mild antipsychotic and antidepressant [1], for the subsequent analysis of its multitarget mechanism of action, the serotonin transporter (SERT) and the serotonin receptor types 3 (5-HT_3_) and 1A (5-HT_1A_) were chosen as target proteins (Table 1).

Figure 2 illustrates the binding mechanisms of L-17, fluoxetine, granisetron, and WAY100135 with the SERT and 5-HT_3_ and 5-HT_1A_ receptors, predicted using LigandScout 4.2.1 software [22].

Figure 3 illustrates the poses of L-17, fluoxetine, granisetron, and WAY100135 within the binding sites of SERT and 5-HT_3_ and 5-HT_1A_ receptors.

### 2.2. Proto-Oncogene Fos Expression

Figure 4 illustrates the expression of proto-oncogene *Fos* in three selected brain areas, including the prefrontal cortex (PFC, A–C), hippocampus (D–F), and amygdala (G–I) of vehicle- and L-17-pretreated rats:

Pretreatment with L-17 decreased c-*Fos* immunoreactivity in the hippocampus and increased it in the amygdala. The expression of c-*Fos* in the PFC was not statistically different between the groups. Summary quantitative assessments were performed from four vehicle- and four L-17-administered rats.

### 2.3. In Vivo Electrophysiology

The mean basal firing activity was 3.46 ± 0.83 Hz. L-17 (0.1–12 mg/kg, i.v.) significantly and dose-dependently (F_8,53_ = 4.84, *p* < 0.001, ANOVA for repeated measures) inhibited the firing activity of 5-HT neurons, reaching a maximal 90% inhibition at 12 mg/kg. WAY100135 partially reversed the L-17-induced inhibition of 5-HT neurons (Figure 5).

## 3. Discussion

In this study, we performed a complex in silico assessment of the pharmacotherapeutic properties of L-17, as well as In Vivo electrophysiological assessment of the effect of this compound on the firing activity of 5-HT neurons in the rat DRN. PASS 10.4 Professional Extended [21] software predicted L-17 to have antidepressant-like properties. This prediction is consistent with the previously reported antidepressant-like effect of L-17 in rats [6]. PASS 10.4 Professional Extended also predicted that the antidepressant-like effect of L-17 can be explained, at least in part, via its 5-HT reuptake inhibition property.

AutoDock Vina 1.1.2 [23] docking energy assessment predicted that L-17 has an affinity to the SERT comparable to fluoxetine, 5-HT_3_ binding affinity similar to granisetron, and 5-HT_1A_ binding affinity comparable to WAY100135. The prediction of the molecular interactions between L-17 and the binding sites of SERT and 5-HT_3_ and 5-HT_1A_ receptors with LigandScout 4.2.1 software [22] revealed that the mechanisms of L-17 binding to the SERT and 5-HT_1A_ receptor are similar to fluoxetine and WAY100135, respectively. L-17 binding to 5-HT_3_ receptors is completely different from that of granisetron. Consistently, the topographic poses of L-17 within SERT and 5-HT_1A_ receptor binding sites are remarkably similar to those of fluoxetine and WAY100135, respectively. Thus, the morpholine cycle nitrogen atom of L-17 is engaged in a p–π exchange with PHE269 of SERT. If protonated, this nitrogen forms a region of undirected electrostatic interaction. Two more structural elements of L-17 participate in five nonspecific hybrid interactions with TYR23, ALA97, ILE100, ALA101, and TYR104 of SERT. Fluoxetine is banded to SERT via the electrostatic interaction of the protonated NH_2_ group. Three additional groups of L-17 form five hydrogen bonds with TYR23, ALA24, ASP26, and SER367x. Three more structural elements participate in six nonspecific hybrid interactions with ILE100, ALA101, TYR104, PHE269, LEU371, and VAL429. L-17 might bind to the 5-HT_3_ receptor by stacking with PHE97. Two structural elements of L-17 participate in four nonspecific hybrid interactions with LEU93, PHE97, VAL100, and LEU123. No stacking is observed in granisetron binding to a 5-HT_3_ receptor. Fixation is exerted by three nonspecific hydrophobic interactions with ILE38, TRP57, and TYR120. The morpholine cycle oxygen atom of L-17 forms a hydrogen bond with TYR366 of the 5-HT_3_ receptor binding site. Two structural elements of L-17 form three nonspecific hydrophobic links with TRP78, PHE88, and ILE89 of the 5-HT_3_ receptor binding site. The protonated nitrogen atom of the morpholine cycle of WAY100135 exerts an electrostatic interaction with TYR366. Two WAY100135 molecule fragments participate in three nonspecific hydrophobic interactions with ALA69, TYR72, TRP78, ILE89, VAL93, ILE165, ALA179, PHE338, and LEU357 of the 5-HT_3_ receptor binding site.

It was demonstrated that pretreatment with L-17 robustly increased c-*Fos* immunoreactivity in the amygdala. The amygdala is a fundamental structure in emotional responses, including fear and anxiety [24]. Thus, it is possible that the modulation of amygdaloid neural excitability is involved in the therapeutic response to antidepressant, anxiolytic, and mood-stabilizing medicines. It has been reported that various CNS drugs, such as typical [12] and atypical antipsychotics [12,13,14], tricyclic antidepressants [13], and SSRIs [13,15,16], increased c-*Fos* immunoreactivity in the amygdala, indicating an increased neural excitability in this brain area. In our previous studies, we found that pyridoindole derivate, SMe1EC2M3, a molecule with putative 5-HT reuptake inhibition properties and antidepressant-like behavioral effect [7], stimulated amygdaloid c-*Fos* immunoreactivity as well [11]. The finding that SSRI-induced amygdaloid c-*Fos* immunoreactivity was potentiated by an antagonist of 5-HT_1A_ autoreceptors indicated that the 5-HT system is involved in the modulation of amygdaloid neural excitability by antidepressant drugs. Thus, it can be summarized that inducement of amygdaloid c-*Fos* immunoreactivity is an important marker for the putative mood-stabilizing and antidepressant-like effect, and the finding that L-17 stimulated proto-oncogene *Fos* expression in the amygdala provides additional support to the hypothesized beneficial effects of this molecule as a future CNS drug.

With regard to proto-oncogene *Fos* expression in other brain areas, we found that acute administration of L-17 significantly decreased c-*Fos* immunoreactivity in the hippocampus, and tended to decrease it in the PFC. It was previously reported that 5-HT depletion led to an increased c-*Fos* expression in these brain areas [20]. Thus, it is possible that the L-17-induced decrease in forebrain c-*Fos* immunoreactivity might be explained by the ability of this drug to elevate extracellular 5-HT concentrations. Further studies should, however, be performed to test this hypothesis.

We found that acute intravenous administration of L-17 significantly and dose-dependently inhibited the firing activity of 5-HT neurons of the DRN. Similar inhibitory effects on 5-HT neuronal firing activity have been observed with other SSRIs, such as citalopram [25], escitalopram [25,26], Wf-516 [27], or paroxetine [28]. Similar to observations of other SSRIs, the L-17-induced inhibition of 5-HT neuronal firing activity was reversed by WAY100135. It is likely that L-17 acts as a potent SERT blocker. Inhibition of SERT by acute levels L-17 leads to an increase in the extracellular 5-HT levels, activation of 5-HT_1A_ autoreceptors, and reduction in the firing activity of 5-HT neurons. The subsequent blockade of 5-HT_1A_ autoreceptors reverses the inhibition of 5-HT neurons.

Unlike escitalopram-induced suppression of 5-HT neuronal firing activity, which was completely reversed by 0.1 mg/kg of WAY100135 [25,26], the L-17-induced inhibition of 5-HT neurons was only partially reversed by WAY100135, even though WAY100135 was administered at 0.3 mg/kg. Thus, it is possible that L-17 interacts with a molecular target (s) other than SERTs. One of these targets predicted by in silico tests is 5-HT receptors. It is possible that L-17 acts, in addition to its function as a SERT blocker, as a partial agonist of 5-HT_1A_ receptors. The competition between L-17 and WAY100135 on 5-HT_1A_ receptors putatively prevents complete WAY100135-mediated recovery of the firing activity of 5-HT neurons. Another possibility might be, however, a direct interaction of L-17 with α_1_-adrenoceptors [29].

In summary, the antidepressant-like properties of L-17 reported in previous studies can be explained, at least in part, by the ability of this compound to modulate central 5-HT neurotransmission. We cannot, however, exclude the interaction of L-17 with receptors modulating the excitability of 5-HT neurons, such as 5-HT_1A_, 5-HT_3_, and/or α_1/2_-adrenergic receptors. It is also possible that L-17 directly interacts with other neurotransmitter systems, such as norepinephrinergic, dopaminergic, and/or histaminergic ones. The L-17-induced modulation of 5-HT transmission is likely to be mediated, at least in part, via the inhibition of 5-HT reuptake and partial antagonism to 5-HT_1A_ receptors. L-17-induced increase in 5-HT transmission putatively results in increased neuronal excitability in the amygdala and decreased neuronal firing activity in the hippocampus. These L-17-induced changes in neural excitability across the brain might be involved in the neuroprotective and antidepressant-like behavioral effects of this compound reported in previous studies. The previously reported antidepressant-like properties of L-17, and its modulatory effect on 5-HT neurons of the DRN and on neural excitability in the amygdala and the hippocampus, reported in this study, suggest that L-17 might be an effective antidepressant drug. Since L-17 also possesses neuro- and cardioprotective properties, it could be useful in affective illness development due to general medical condition, such as post-stroke and post-MI depression. It should be also stated that the combined in silico (computerized prediction of molecular binding mechanisms), Ex Vivo (assessment of neuronal excitability using c-*Fos* immunoreactivity measurement), and In Vivo (direct assessment of the excitability of monoamine-secreting neurons) investigation might be a useful strategy for early preclinical assessment of the efficacy and neural mechanism of action of novel CNS drugs. In this specific study, the usage of in silico prediction techniques allowed us to forecast the SERT- and 5-HT_1A/3_-receptor-mediated antidepressant effect of L-17. The predicted 5-HT reuptake inhibitory effect of L-17 was confirmed by In Vivo electrophysiology, and the putative 5-HT-mediated effect on neural excitability in the amygdala and hippocampus by the c-*Fos* expression assessment.

## 4. Materials and Methods

### 4.1. In Silico Predictions

Using PASS 10.4 Professional Extended software (Institute of Biomedical Chemistry, Moscow, Russia) [21], the presence (Pa) or absence (Pi) probability of 480 systemic types of pharmacotherapeutic activity was calculated. Promising activities were those with Pa ≥ 0.1 and Pa/Pi ≥ 1.0.

Furthermore, for a comparative evaluation of the L-17 compound’s affinity to the selected biotargets, the docking of the compound to the specific binding sites of these proteins was performed. Five experimental X-ray 3D models of SERT were obtained from Protein Data Bank in Europe [30]. Among these five models, the longest one (PDB code 5I6X), with the maximum resolution including an inhibitor, was chosen to allow the unambiguous determination of the binding site position [31]. The experimental 3D models for 5-HT_3_ and 5-HT_1A_ receptors were not available from Protein Data Bank in Europe; therefore, a search for the best theoretical 3D models from the Database of Comparative Protein Structure Models [32] was conducted. Among the available models, the longest ones, with the highest statistical significance, were selected for 5-HT_3_ [33] and 5-HT_1A_ [34] receptors.

The 3D models of L-17 and reference molecules were constructed using the molecular mechanics methods in MarvinSketch15.6.15 software (Chemaxon, Budapest, Hungary) [35], followed by optimization with the semi-empirical quantum chemical method PM7 in MOPAC2016 software (James Stewart, Colorado Springs, CO, USA) [36]. Granisetron, WAY100135, and fluoxetine were chosen as reference molecules since they are well-known as blockers of 5-HT_3_ and 5-HT_1A_ receptors and SERTs, respectively. Fluoxetine is also one of the most widely used antidepressant drugs. The docking was performed using AutoDock Vina 1.1.2 [23], five times for each compound into each target, and then the spectrum of energies was used to determine their minimum binding energies. To reveal the mechanisms of serotoninergic action of the L-17 compound, an analysis of its affinity spectrum in comparison with the affinity spectra of reference compounds was performed. The primary information about the reference compounds for the target proteins was obtained from the UniProtKB database [37]. For each reference compound found in UniProtKB, the mechanism of its action was clarified using DrugBank [38]. In the case that information in UniProtKB and DrugBank datasets was not sufficient, the search for references was performed in the IUPHAR database of pharmacologically relevant proteins and their ligands [39]. The SSRI fluoxetine and the selective antagonists of 5-HT_3_ (granisetron) and 5-HT_1A_ (WAY100135) receptors were used as reference molecules. Using the obtained energy spectra of ΔE, the pK values were calculated using the formula:$$pK=-\mathrm{lg}e^{-\Delta E/RT}$$
where *R* is the universal gas constant (8.314 J × mol^−1^ × K^−1^) and *T* is the temperature, set as 300 K. The molecular binding mechanism of L-17, granisetron, and WAY100135 with SERT and with 5-HT_3_ and 5-HT_1A_ receptors was further evaluated using LigandScout 4.2.1 software [22]. The data on binding sites were found in the available literature. For SERT, the key amino acids of the binding site are Gly94, Ala96, Val97, Asn101, Ser336, Asn368, Leu434, Asp437, and Ser438 [31]. For the 5-HT_3_ receptor, the key amino acids of the binding site are Tyr229, Phe221, Asn123, Trp85, Trp178, Tyr148, Arg87, Gln146, and Tyr138 [33]. For the 5HT_1A_ receptor, the key amino acids of the binding site are Tyr56, Gln57, Asp76, Val77, Ser159, Trp318, Phe321, Phe322, Thr339, Gly342, Ala343, Ile345, and Asn346 [34].

### 4.2. Assessment of c-Fos Immunoreactivity

For the assessment of the proto-oncogene c-*Fos* expression, rats were pretreated with L-17 (60 mg/kg, intraperitoneally, i.p.) twenty-four hours and one hour before immunoreactivity measurements. Expression of c-*Fos* was analyzed, as we reported previously [11,40]. The rats were anesthetized by a combined treatment of zoletil (30 mg/kg, Virbac, Carros, France) and xylariem (15 mg/kg, Riemser Pharma GmbH, Greifswald, Germany) in volumes of 0.1 mL and 0.24 mL/300 g b.w., respectively. Then, they were perfused transcardially with 50 mL of cold isotonic saline containing 150 µL of heparin (5000 IU/mL, Zentiva, Hlohovec, Slovakia) followed by 200 mL of fixative containing 4% paraformaldehyde (Sigma-Aldrich, Germany) in 0.1 M phosphate buffer (PB, pH 7.4). Immediately after perfusion, the brains were removed from the skulls, postfixed in a fresh fixative overnight, washed in 0.1 M PB at 4 °C overnight, and infiltrated with 30% sucrose (Slavus, Podunajské Biskupice, Slovakia) at 4 °C for 48 h. Coronal sections of 35 µm thickness were alternatively cut over the whole brain in a cryocut (Hyrax C-50). The sections were repeatedly washed in 0.1 M PB and pretreated with 0.3% H_2_O_2_ (Sigma-Aldrich, Munich, Germany) in 0.1 M PB for 10 min at room temperature (RT). Thereafter, the sections were rinsed 3 × 10 min in 0.1 M PB and exposed to rabbit anti-*Fos* polyclonal antibody (1:2000) in 0.1 M PB containing 4% normal goat serum (Gibco, Grand Island, NY, USA), 0.5% Triton X-100 (Sigma-Aldrich, Munich, Germany), and 0.1% sodium azide (Sigma-Aldrich, Munich, Germany) for 48 h at RT. After several rinses in PB, the sections were incubated with biotinylated goat anti-rabbit IgG (1:500, BA-9200, VectorStain Elite ABC Kit, Vector Laboratories, Burlingame, CA, USA) in PB for 90 min at RT. Next, the PB rinsing was followed by incubation with the avidin–biotin–peroxidase complex (1:250) for 90 min at RT. After several washes in 0.05 M sodium acetate buffer (SAB, pH 6.0), c-*Fos*-antigenic sites were visualized by 3,3′-diaminobenzidine tetrahydrochloride (0.0625% DAB) enhanced with 2.5% nickel chloride (Sigma-Aldrich, Munich, Germany), in SAB containing 0.0006% H_2_O_2_. Developing time was 6–8 min. The heavy metal intensification of DAB yielded black staining of c-*Fos*-labeled nuclei.

The topographic mapping of the c-*Fos* protein immunolabeled cells was performed in three brain areas (PFC, hippocampus, and central nucleus of the amygdala), identified based on the rat brain atlas [11,41]. The individual structures investigated were captured unilaterally from 6–10 representative sections using an Axio Imager A1 light microscope (Carl Zeiss, Jena, Germany) coupled to a video camera and monitor. Since the staining of each individual *Fos*-immunoreactive cell varied from very intense to very light, c-*Fos* profile counting was performed on pictures inverted by the Adobe Photoshop 7.0 “invert” adjustment to achieve white c-*Fos* profiles on a black background. The counting of the c-*Fos* profiles was performed manually on a PC computer by a person blind to the design of the experiment.

### 4.3. In Vivo Electrophysiology

In Vivo electrophysiological assessment of the excitability of 5-HT neurons of the DRN was performed as previously described [7,26,42,43]. Adult male Wistar rats, weighing 300–350 g, were ordered from the Animal Breeding Facility of The Institute of Experimental Pharmacology and Toxicology, Center of Experimental Medicine, Slovak Academy of Sciences in Dobrá Voda, Slovakia. Rats were anesthetized with chloral hydrate (0.4 g/kg, intraperitoneally, i.p., Lambda Life s.r.o., Bratislava, Slovakia;) and maintained in a stereotaxic frame (David Kopf Instruments, Tujunga, CA, USA). Rat body temperature was maintained between 36 and 37 °C with a heating pad (Gaymor Instruments, Orchard Park, NY, USA). The scalp was opened, and a 3 mm hole was drilled in the skull for insertion of electrodes. Glass pipettes were pulled with a DMZ Universal Electrode Puller (Zeitz-Instruments GmbH, Martinsried, Germany) to a fine tip approximately 1 μm in diameter and filled with 2M NaCl solution. Electrode impedance ranged from 4 to 6 MΩ. The pipettes were lowered into the DRN (7.8–8.3 mm posterior to bregma and 4.5–7.0 mm ventral to brain surface) [41] with a David Kopf Instruments hydraulic micro-positioner. The action potentials generated by 5-HT neurons were recorded using the ADInstruments Extracellular Recording System (Dunedin, New Zealand). The 5-HT neurons were identified by bi- or tri-phasic action potentials with a rising phase of long duration and regular firing rate of 0.5–5.0 Hz [44]. All experimental procedures were approved by the Animal Health and Animal Welfare Division of the State Veterinary and Food Administration of the Slovak Republic (Permit number Ro 3592/15-221) and conformed to Directive 2010/63/EU of the European Parliament and the Council on the Protection of Animals Used for Scientific Purposes.

## Figures and Tables

**Figure 1 ijms-22-13626-f001:**
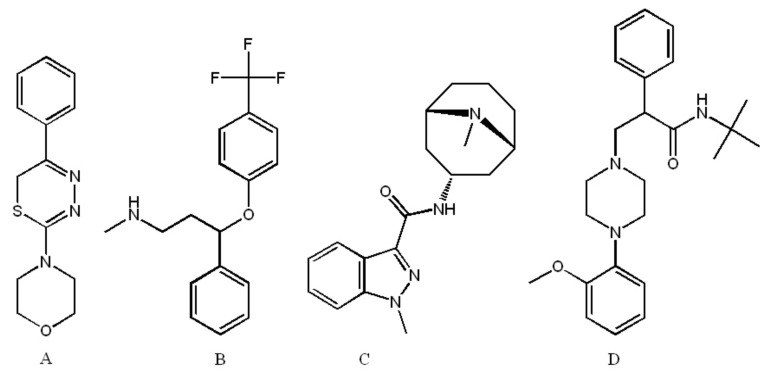
Structural chemical formulas of 2-morpholino-5-phenyl-6H-1,3,4-thiadiazine, hydrobromide: (L-17) (**A**), the SSRI fluoxetine (**B**), 5-HT_3_ receptor antagonist granisetron (**C**), and 5-HT_1A_ receptor antagonist WAY100135 (**D**).

**Figure 2 ijms-22-13626-f002:**
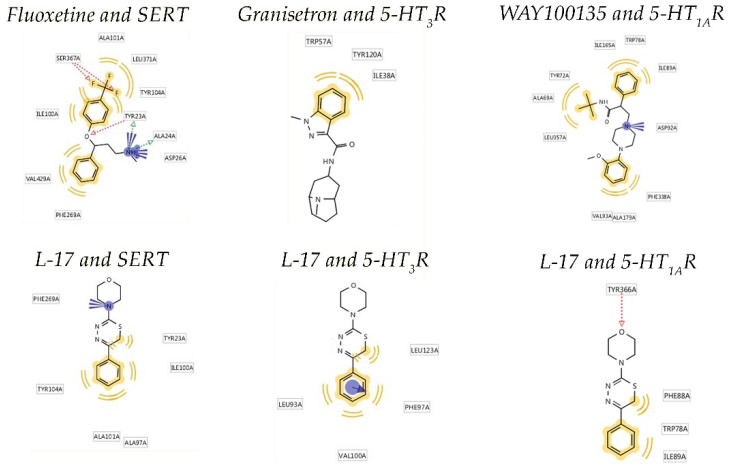
Binding of L-17, fluoxetine, granisetron, and WAY100135 to the SERT, and 5-HT_3_ (5-HT_3_R) and 5-HT_1A_ (5-HT_1A_R) receptors.

**Figure 3 ijms-22-13626-f003:**
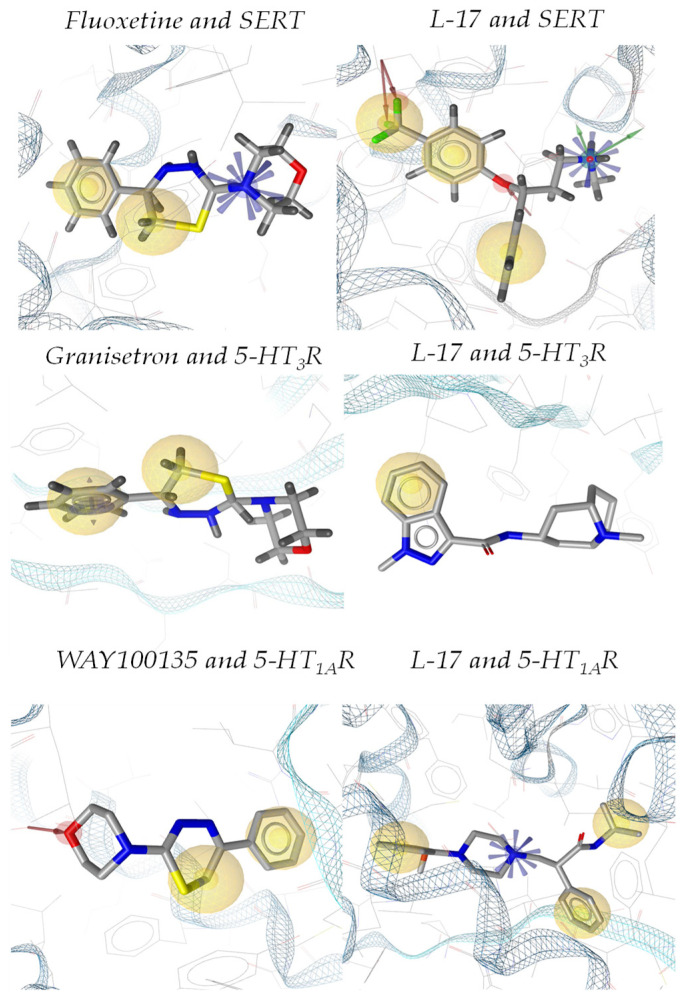
Poses of L-17, fluoxetine, granisetron, and WAY100135 within the binding sites of SERT, and 5-HT_3_ and 5-HT_1A_ receptors. The pose of L-17 within the SERT binding site is remarkably close to that of fluoxetine. There is also a partial similarity between L-17 and WAY100134 poses within the binding site of the 5-HT_1A_ receptor. The poses of L-17 and granisetron within the binding site of the 5-HT_3_ receptor are remarkably different.

**Figure 4 ijms-22-13626-f004:**
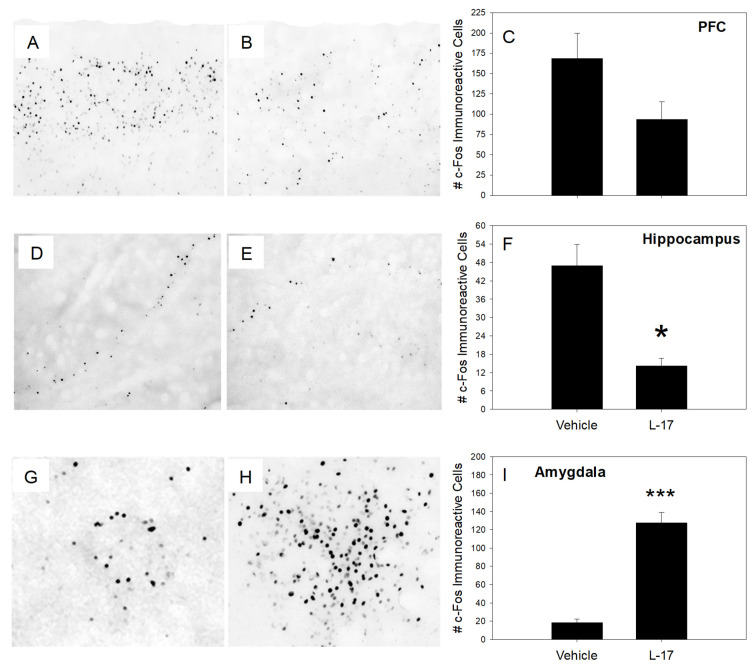
Representative sections illustrating the expression and distribution of c-*Fos* protein immunolabeled cells in the prefrontal cortex PFC, (**A–C**), hippocampus (**D**–**F**), and the central nucleus of the amygdala (**G**–**I**) in vehicle- and L-17-pretreated rats. * *p* < 0.05 and *** *p* < 0.001, two-tailed Student’s *t*-test.

**Figure 5 ijms-22-13626-f005:**
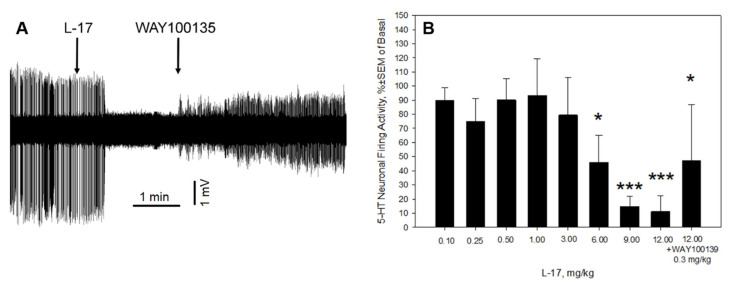
L-17 significantly and dose-dependently inhibited the firing activity of 5-HT neurons. (**A**): representative recording from a 5-HT neuron during L-17 (12 mg/kg) and WAY100135 (0.3 mg/kg) administration. (**B**): Summary effect of L-17 (0.1–12 mg/kg) and WAY100135 (0.3 mg/kg) on the spontaneous firing activity of 5-HT neurons of the DRN (data from 8 neurons from 7 rats).* *p* < 0.05 and *** *p* < 0.001, Bonferroni post-hoc test.

**Table 1 ijms-22-13626-t001:** Docking energy (ΔE), binding affinity (pK), and relevant affinity (RA) of L-17 to the SERT and 5-HT_3_/5-HT_1A_ receptors, compared to the SSRIs fluoxetine, granisetron, and WAY100135.

Target	SERT			5-HT_3_			5-HT_1A_		
Molecule	ΔE, kcal/mol	pK	RA	ΔE, kcal/mol	pK	RA	ΔE, kcal/mol	pK	RA
L-17	−8.1	5.90	0.87	−6.6	4.81	0.94	−7.9	5.78	0.85
Fluoxetine	−9.3	6.78	-	-	-	-	-	-	-
Granisetron	-	-	-	−7.0	5.10	-	-	-	-
WAY100135	-	-	-	-	-	-	−9.3	6.78	-

## Data Availability

The original research data is avaliable upon requet.

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
