# Peer review of "Combined In Silico, Ex Vivo, and In Vivo Assessment of L-17, a Thiadiazine Derivative with Putative Neuro- and Cardioprotective and Antidepressant Effects"

_ijms, 2021, doi:10.3390/ijms222413626_

Round 1
Reviewer 1 Report
The submitted paper discusses the results of research on the mechanism of action of L-17 compound, thiadiazine derivative with neuro and cardioprotective and antidepressant effects. The topic is interesting, and the choice of the compounds tested was confirmed in the earlier works of the authors. Earlier studies conducted on animal models indicated a possible therapeutic effect of L-17 in the case of myocardial infarction and pancreatitis, as well as an immunostimulatory, neuroprotective and antidepressant effect. The presentation of the research in the current work is conducted in a systematic way, discussing own results against the background of the works of other researchers. This time it concerns experiments (in silico, ex vivo and in vivo) on the mechanism of the likely beneficial effect of the L-17 compound on the central nervous system. Additionally, it can be seen that presented research strategy (first computer prediction of binding mechanisms, then measurement of immunoreactivity, and finally direct evaluation of neuronal excitability in rats) may be a useful tool for early assessment of the affectivity and neuronal mechanism of action for new drugs. The only remark concerns the lack of a separate summary of the research results (Conclusions) at the end of the manuscript.
Author Response
The changes made in the revised version of the manuscript are highlighted using the track changes mode and by red colour in this letter.
The submitted paper discusses the results of research on the mechanism of action of L-17 compound, thiadiazine derivative with neuro and cardioprotective and antidepressant effects. The topic is interesting, and the choice of the compounds tested was confirmed in the earlier works of the authors. Earlier studies conducted on animal models indicated a possible therapeutic effect of L-17 in the case of myocardial infarction and pancreatitis, as well as an immunostimulatory, neuroprotective and antidepressant effect. The presentation of the research in the current work is conducted in a systematic way, discussing own results against the background of the works of other researchers. This time it concerns experiments (in silico, ex vivo and in vivo) on the mechanism of the likely beneficial effect of the L-17 compound on the central nervous system. Additionally, it can be seen that presented research strategy (first computer prediction of binding mechanisms, then measurement of immunoreactivity, and finally direct evaluation of neuronal excitability in rats) may be a useful tool for early assessment of the affectivity and neuronal mechanism of action for new drugs.
We thank the reviewer for these motivating comments
The only remark concerns the lack of a separate summary of the research results (Conclusions) at the end of the manuscript.
We agree with the reviewer that a separate summary sentence is necessary. It was added as follows:
… the combined in silico (computerized molecular binding mechanisms prediction), ex vivo (assessment of the neuronal excitability using c-Fos immunoreactivity measurement) and in vivo (direct assessment of excitability of monoamine-secreting neurons) investigation might be a useful strategy for early preclinical assessment of the affectivity and neural mechanism of action in the novel CNS drugs. In this specific study, the usage of in silico prediction techniques allowed to forecast SERT and 5-HT1A/3-receptor mediated antidepressant effect of L-17. The predicted 5-HT reuptake inhibitory effect of L-17 was confirmed by in vivo electrophysiology, and putative 5-HT-mediated effect on neural excitability in the amygdala by the c-Fos expression assessment.

Reviewer 2 Report
Row 270 "The 3D models of L-17 and reference molecules were constructed using the molecular ...". reference molecules are drugs used to treat different pathological condition and that are used in therapy?
Which experiments highlight cardioprotective effects of the compound that has been synthesized?
Author Response
The changes made in the revised version of the manuscript are highlighted using the track changes mode and by red colour in this letter.
Row 270 "The 3D models of L-17 and reference molecules were constructed using the molecular ...". reference molecules are drugs used to treat different pathological condition and that are used in therapy?
We thank thew reviewer for this inquiry. This point was clarified:
… 3D models of L-17 and reference molecules were constructed using the molecular mechanics' methods in the MarvinSketch15.6.15software [35], followed by optimization with the semi-empirical quantum chemical method PM7 in the MOPAC2016 software [36]. Granisertone, WAY100135, and fluoxetine were chosen as reference molecules since they are well known as blockers of 5-HT3 and 5-HT1A receptors and SERTs, respectively. Fluoxetine is one also one of the of the most widely used antidepressant drugs.
Which experiments highlight cardioprotective effects of the compound that has been synthesized?
We agree with the reviewer that this important point needs clarification. The following text wad added:
… we have shown, using animal models, a putative therapeutic effect of L-17 in myocardial infarction (MI) [2-4] and pancreatitis [5] in animal models. Thus, L-17 significantly decreased the initial infection area and accelerated the granulocytotic and supressed cytokine component of the inflammatory reaction in rats after the left coronary artery coagulation [2,3].
Reviewer 3 Report
Paper entitled:” Combined in silico, ex vivo, and in vivo Assessment of L-17, a Thiadiazine Derivative with Putative Neuro- and Cardioprotective and Antidepressant Effects” (ijms- 1456329) presents an interesting study about the mechanism of action 2-morpholino-5-phenyl-6H-1,3,4-thiadiazine, hydrobromide (L-17), with putative neuro- and cardioprotective and antidepressant-like effects. The presented results of the study may contribute to the development of a therapy for early preclinical assessment of the affectivity and neural mechanism of action in the novel CNS drugs. In my opinion this work is largely hypothetical but could contribute to the development of further strategic research towards the treatment of neurological diseases. However, there are some issues which should be addressed before publication in IJMS.
- L78: please expand the SSRI abbreviation
- There are some fuzzy text next to Fluoxetine in Figure 2 that needs to be corrected
- There are some fuzzy text next to L-17 &SERT in Figure 2 that needs to be corrected
- L-17 &5-HT1AR - please correct the clarity of the subtitles in Fig 2
- In Figure 4c there is no description on the X axis, please complete this
Author Response
The changes made in the revised version of the manuscript are highlighted using the track changes mode and by red colour in this letter.
Paper entitled:” Combined in silico, ex vivo, and in vivo Assessment of L-17, a Thiadiazine Derivative with Putative Neuro- and Cardioprotective and Antidepressant Effects” (ijms- 1456329) presents an interesting study about the mechanism of action 2-morpholino-5-phenyl-6H-1,3,4-thiadiazine, hydrobromide (L-17), with putative neuro- and cardioprotective and antidepressant-like effects. The presented results of the study may contribute to the development of a therapy for early preclinical assessment of the affectivity and neural mechanism of action in the novel CNS drugs. In my opinion this work is largely hypothetical but could contribute to the development of further strategic research towards the treatment of neurological diseases. However, there are some issues which should be addressed before publication in IJMS.
We thank the reviewer for these motivating comments
L78: please expand the SSRI abbreviation
The abbreviation is now expanded:
such as typical [12] and atypical antipsychotics [12-14], tricyclic antidepressants [13], and selective serotonin reuptake inhibitors (SSRIs) [13, 15, 16].
There are some fuzzy text next to Fluoxetine in Figure 2 that needs to be corrected
There are some fuzzy text next to L-17 &SERT in Figure 2 that needs to be corrected
L-17 &5-HT1AR - please correct the clarity of the subtitles in Fig 2
The text labels in the Figure 2 are now corrected
In Figure 4c there is no description on the X axis, please complete this
The axes description is now clarified